# An Infusion Containers Detection Method Based on YOLOv4 with Enhanced Image Feature Fusion

**DOI:** 10.3390/e25020275

**Published:** 2023-02-02

**Authors:** Lei Ju, Xueyu Zou, Xinjun Zhang, Xifa Xiong, Xuxun Liu, Luoyu Zhou

**Affiliations:** 1College of Electronic and Information Engineering, Yangtze University, Jingzhou 434023, China; 2College of Electronic and Information Engineering, South China University of Technology, Guangzhou 510641, China

**Keywords:** object detection, YOLOv4, artificial intelligence, feature information

## Abstract

The detection of infusion containers is highly conducive to reducing the workload of medical staff. However, when applied in complex environments, the current detection solutions cannot satisfy the high demands for clinical requirements. In this paper, we address this problem by proposing a novel method for the detection of infusion containers that is based on the conventional method, You Only Look Once version 4 (YOLOv4). First, the coordinate attention module is added after the backbone to improve the perception of direction and location information by the network. Then, we build the cross stage partial–spatial pyramid pooling (CSP-SPP) module to replace the spatial pyramid pooling (SPP) module, which allows the input information features to be reused. In addition, the adaptively spatial feature fusion (ASFF) module is added after the original feature fusion module, path aggregation network (PANet), to facilitate the fusion of feature maps at different scales for more complete feature information. Finally, EIoU is used as a loss function to solve the anchor frame aspect ratio problem, and this improvement allows for more stable and accurate information of the anchor aspect when calculating losses. The experimental results demonstrate the advantages of our method in terms of recall, timeliness, and mean average precision (mAP).

## 1. Introduction

Intravenous input is a very important treatment while infusion bottles and infusion bags are the most common containers for infusion. With the spread of the epidemic in recent years, many patients requiring infusions have gathered in hospitals, putting increasing pressure on the healthcare system. Detecting infusion containers can help healthcare professionals better understand the status of infusion recipients, which makes planning easier and reduces the stress of medical staff.

The detection of infusion containers plays an important role in reducing the pressure on medical personnel and is essentially part of the field of object detection [1]. There are traditional methods and deep learning in the field of object detection. With the development of deep learning, even in some specific areas traditional methods are as effective as deep learning, and the scalability and robustness of deep learning is making it increasingly mainstream. In hospital scenarios, individual infusion containers often overlap and are not at the same distance from the camera, which makes their detection difficult. In order to solve this problem here, we modify the neck part of the YOLO architecture by adding coordinate attention (CA) [2] after the backbone to effectively capture location and channel information, improve SPP to CSP-SPP to enhance the ability of feature fusion [3], and add the feature fusion module ASFF [4] after PANet to increase the depth of the network; this improved model is named NMYOLO. In addition, we adopt EIoU [5] instead of CIoU in the loss function to solve the problem of the ambiguous aspect ratio of anchor frame, resulting in more effective detection while maintaining the inference speed.

The main contributions of this paper are summarized as follows: (1)The neck part of YOLOv4 [6] is improved by using replacing the original modules with several more effective modules. We have improved SPP to CSP-SPP, which enhanced the feature extraction capability of the model. We have also added CA and ASFF to obtain more image information. These improvements validate the scalability of the YOLO architecture and also lay the foundation for further research.(2)The loss function of YOLOv4 is improved by replacing CIOU with EIOU in calculating the width-to-height ratio of the anchor box. This not only results in more stable and accurate prediction of boxes but also reduces the training time and calculation cost.

The content in this paper is structured as follows. Presented in Section 2 is the related work of our study. Section 3 describes the methods of YOLOv4 and Section 4 introduces the details of NMYOLO. A comparison between the specific parameters of the experiment and the final results is presented in Section 5. Finally, conclusions are drawn in Section 6.

## 2. Related Work

Target detection is the main component of computer vision. In this section, we review the solutions in deep learning for object detection.

Common deep learning object detection algorithms can be roughly divided into two categories: the first is the two-stage [7] detectors, which are also known as object detection models based on the candidate region proposal. The process of object detection involves, first, the generation of candidate regions on the image and extraction of the corresponding image features, which are then input into the classifier for judgment. Region-based convolutional neural network (RCNN), as reported by Grishick in 2014 [8], is the first of its kind in terms of two-stage detectors. On the basis of the original detector, Grishek proposed fast R-CNN [9] and faster R-CNN [10,11], both of which significantly reduced the time consumed by algorithmic reasoning with improved accuracy. The second categories is the one-stage [7] detectors, which began with You Only Look Once (YOLO) [12,13] proposed by Redmon et al. in 2016 and has gradually become the mainstream object detection algorithm after several years of development. Common one-stage detectors include YOLO series and the single shot multi-box detector (SSD) [14] proposed by Liu. One-stage detectors do not have the step of generating candidate box regions, so are much faster than two-stage detectors in inference, which allows the computational overhead to be reduced.

In recent years, the self-attention mechanism has also been widely used in target detection. In 2017, Vaswani et al. proposed transformer [15], a model that demonstrates that self-attention is very effective in deep learning. Wang et al. proposed the no-local network [16], a model that can capture long-range dependencies more easily. Hu et al. proposed SENet [17] in 2018, a module that can be easily added to other models and improve accuracy, which triggered the thinking about self-attention in the field of vision. Immediately after, Woo et al. proposed CBMA [18], a lightweight module that can also be easily integrated into other CNN architectures, which is divided into channel blocks and spatial blocks that can be used to regenerate feature maps.

## 3. The Methods of YOLOv4

### 3.1. CSP Structure

Cross stage partial network (CSPNet) [19] is characterized by the integration of feature mapping at the beginning and end of the network stages. Figure 1 shows the application of CSPNet on the ResNe(X)t [20] network structure.

BaseLayer is the feature extraction layer in Figure 1, and the extracted feature information is divided into two parts, where the output of Part 2 is processed into a Res(X) module and the output feature map is spliced with the feature map of Part 1 to obtain the final output. This can reduce the amount of duplicated feature information in the network through cross-stage connection and, at the same time, improve the learning ability of the network to enhance the final result. This is the reason why both YOLOv4 and YOLOv5 choose the CSP-structured network as the backbone feature extraction network.

### 3.2. YOLOv4

YOLOv4 is an object detection model proposed by Bochkovskiy et al. in 2020 containing many improvements from YOLOv3. It includes the use of the mosaic data enhancement method, which solves the problem of difficult detection of small targets. The idea of CPSNet is absorbed to replace the backbone from Darknet53 to CSPDarknet53, and the cross-stage residual connection structure is used to obtain more effective feature extraction. The activation function of the backbone is replaced from Leakey_relu to Mish, while SPP [21] and PANet [22] are used in the feature pyramid module instead of feature pyramid networks (FPN) [23]. The overall structure for an input image size of 416 × 416 is shown in Figure 2.

After the input image passes through the backbone feature extraction network, three different scales of 52 × 52, 26 × 26 and 13 × 13 features are output to the feature fusion layer, where the 13 × 13 scale features are enhanced by SPP module and then fused with the 52 × 52 and 26 × 26 scale features in PANet after upsampling and downsampling feature fusion concatenation. The features at different scales are extracted several times to obtain a better fusion effect. Finally, the fused features of different scales are input into three YOLO heads for prediction. In addition, YOLOv4 uses CIoU loss of bound box, which can be described as
(1)$$\mathrm{CIoU}=\mathrm{IoU}-\frac{\rho^{2}{(b,b}^{gt})}{d^{2}}-\alpha \nu$$
where *ρ*(*b*,*b*^gt^) is the distance between the prediction box and the center point of the ground truth box, *d* is the diagonal distance of the smallest rectangular box containing both the real and prediction boxes, and *α* and *v* are calculated as follows:(2)$$\alpha =\frac{\nu }{1-\mathrm{IoU}+\nu }$$
(3)$$\nu =\frac{4}{\pi^{2}}{(\arctan\frac{w^{gt}}{h^{gt}}-\arctan\frac{w}{h})}^{2}$$
where *w*^gt^ and *h*^gt^ represent the width and height of the ground truth box, respectively, while *w* and *h* represent the width and height of the prediction box, respectively.

## 4. The Detail of NMYOLO

Currently, CSPDarknet53, which is used in the overall structure of YOLOv4 to extract backbone network features, demonstrates good performance in object detection tasks, so it is retained in our proposed model. In this study, we have mainly improved the neck module, as most of the fusion processing for feature information is located in this module.

### 4.1. ASFF

Feature fusion is a very important component of the target detection task because the fusion of different-scale features is an important for improving the performance of model detection. Therefore, we choose to add a feature fusion module after PANet to improve the depth and detection ability of the model.

The structure of ASFF is shown in Figure 3. In our model, ASFF is added to PANet. The 52 × 52, 26 × 26 and 13 × 13 scales of feature maps are fused with other scales and then input to YOLO heads for prediction. In the process of feature fusion, ASFF uses the weight parameter to control the contribution of different feature maps, which also reflects the idea of attention, so it can help the network to better fuse the extracted high-level information and low-level information and thus improve the final detection capability.

### 4.2. Coordinate Attention

In the task of object detection, the effect of the improved attention mechanism on the final result is obvious. CA can capture the direction and position perception information while capturing the cross-channel information by embedding the position information in the picture into the channel attention. Therefore, CA can help the model to locate it more accurately and identify the target in the picture. At the same time, CA can improve the effect without occupying excessive computational overhead because it is a lightweight module. The structure of CA is shown in Figure 4.

In CA, the input information is first passed through a residual structure, and the attentional feature information is subsequently and separately extracted according to the horizontal and vertical directions. Thus, the key *X*-axis and *Y*-axis position information of the input feature map is obtained. The formula is as follows:(4)$$Z_{c}^{h}(h)=\frac{1}{W}\underset{0\leq i<\;W}{\sum }x_{c}\left(h,i\right)$$
(5)$$Z_{c}^{w}(w)=\frac{1}{H}\underset{0\leq j<\;H}{\sum }x_{c}\left(j,w\right)$$
where *x*_c_(*h*,*c*) represents the *c*-th input channel of height h, which is output Z*_c_^h^*(*h*) after being encoded with a convolutional kernel of size (1,*W*). *x_c_*(*j*,*w*) represents the *c*-th input channel of width *w*, which is output as Z*_c_^w^*(*w*) after being encoded by a convolutional kernel of size (1,*H*). Then, (4) and (5) are stitched together and fed into a convolutional module of 1 × 1, and the nonlinear data are then obtained through the activation function and then divided into two different sets of feature plots, which are defined as follows:(6)$${f=\delta (F}_{1}{([z}^{h}{,z}^{w}]))$$In (6), *F*_1_ is a convolution of 1 × 1, while *δ* is a nonlinear activation step. After that, the output is separately fed into a convolutional module of 1 × 1, and the sigmoid is then used to gain the weight of attention.
(7)$$g^{h}{=\sigma (F}_{h}{(f}^{h}))$$
(8)$$g^{w}{=\sigma (F}_{w}{(f}^{w}))$$In (7) and (8), *f^h^* and *f^w^* are the outputs of the previous step, *F_h_* and *F_w_* represent the corresponding 1 × 1 convolution, and σ is the sigmoid activation function. In addition, *g_c_^h^*(*i*) and *g_c_^w^*(*j*) are used as attention weights. Finally, the coordinate attention output features obtained after multiplying the initially input data with the horizontal and vertical weights are multiplied, and the final result can be written as
(9)$$y_{c}\left(i,j\right)=x_{c}{(i,j)\times g}_{c}^{h}{(i)\times g}_{c}^{w}\left(j\right)$$

### 4.3. Improvements of SPP

Inspired by the CSP structure of the backbone network, the SPP structure after the backbone network is changed to a CSP-SPP structure in our model to better capture and fuse the featured information of the images. The SPP structure and the changed structure are shown in Figure 5.

As shown in Figure 5, in the CSP-SPP structure, the output of the 1 × 1 convolutional module is divided into two parts before SPP, one of which enters the SPP structure normally, and then outputs after the convolutional module of 1 × 1, 3 × 3, and 1 × 1. The other part conducts concatenation with its output. This method can be used to map and connect the characteristics of different stages through cross-stage connections, effectively strengthening the learning ability of the network.

### 4.4. The Structure of NMYOLO

Based on the above improvements, we proposed NMYOLO. The final overall model also includes three parts: backbone, neck, and head. The overall network structure of the model is shown in Figure 6.

In NMYOLO, the size of input image is 416 × 416, and three feature maps of different sizes are generated after backbone, which first send to CA for processing to obtain positional attention information in different directions, thereby further improving the feature extraction ability of the main target of the network. Then, the feature map of 13 × 13 scale is fed into CSP-SPP for processing, and the 13 × 13 scale feature map is captured and first fused with feature information.

Following this, the feature map of three scales is sent to PANet for feature fusion, and information feature extraction of different scales is realized, and ASFF processing is then carried out such that a group of features can contain information after the fusion of other scale feature maps, and the feature maps of different scales finally obtained by the fusion are input to YOLO head for output. Because the final output scale sizes are 52 × 52, 26 × 26, and 13 × 13, NMYOLO demonstrates good performance in detecting targets of different scale sizes.

In addition, unlike the prediction box loss function CIoU in YOLOv4, the loss function used by NMYOLO for prediction box classification is EIoU. With the same consideration of the overlapping area of the bounding box and the distance of the center point, treatment of the aspect ratio by EIoU involves calculating the true difference between the individual width and height and its confidence, while CIoU only calculates the difference between its overall aspect ratio. EIoU takes a more comprehensive view and is able to obtain more stable and accurate anchor frame information to speed up training and improve detection results. EIoU is shown in (10):(10)$$\begin{array}{c} L_{\mathrm{EIoU}}=L_{\mathrm{IoU}}+L_{dis}+L_{asp}\\ =1-\mathrm{IoU}+\frac{p^{2}(b,b^{gt})}{d^{2}}+\frac{p^{2}(w,w^{gt})}{w_{\min}^{2}}+\frac{p^{2}(h,h^{gt})}{h_{\min}^{2}} \end{array}$$
where IoU is the ratio of intersection and union between prediction box and true box. In addition, *ρ*(*w*,*w*^gt^) and *ρ*(*h*,*h*^gt^) represent the distance between the predicted and true width–height center point, while *w*_min_ and *h*_min_ are the width and height of the minimum add-in box that covers both the prediction box and the true box.

### 4.5. Evaluation Metrics

We divided the model evaluation into subjective and objective evaluation metrics. For subjective evaluation, the detection effect graph of each model is output, with observation of its detection effect and whether there is any wrong or missing detection. For objective evaluation, recall, mean average precision (mAP), giga floating point of operations (GFLOPs), and frame per second (FPS) of each model are used as evaluation indicators.

Among them, the GFLOPs indicator is used to measure the model complexity, and recall and mAP calculation are as follows: (11)$$\mathrm{Recall}=\frac{\mathrm{TP}}{P}$$
(12)$$\mathrm{mAP}=\frac{\sum_{q=1}^{n}{\mathrm{AP}}_{q}}{n}$$

Equation (11) is the recall rate calculation formula, where TP indicates the number of positive samples that are correctly predicted as positive. *P* represents the number of samples in which all predictions are positive. Equation (12) is the formula for mAP calculation, with n representing the total number of categories, and AP*q* representing the average precision of class *q*. The mAP values at IoU thresholds of 0.5 and 0.75 are usually denoted by mAP50 and mAP75, respectively, and mAP0.5:0.95 is used to characterize the statistical average of mAP IoU thresholds starting from 0.5 and increasing sequentially by 0.05 up to 0.95.

## 5. Experiments and Results

### 5.1. Dataset Preparation

There are no datasets on infusion bottles and infusion bags available online, so we took initiative in establishing such a dataset [24]. In this study, a total of 9959 pictures of infusion bottles and infusion bags were taken, collected, and sorted. These include images of single infusion containers and images of multiple infusion containers overlapping each other, while some of these images were taken by adjusting the camera aperture light and dark to simulate environmental changes, and distracting factors such as transparent glasses and common water glasses for drinks were added to some other images. After completing the basic data acquisition and labeling, we apply random masking to a portion of the images. This increases the complexity of our dataset and will allow us to more thoroughly evaluate the effect of the tested models. There are 7661 images for training, 852 images for validation, and 946 images for testing. After obtaining the dataset images, we used Labelimg for annotation. There are five classes in our dataset, inf_bot and inf_bag are infusion bottles and infusion bags, respectively, while bot, sprite, and cola are interference classes used to enhance robustness. The details are shown in Figure 7 and Figure 8, while some examples of the dataset images are shown in Figure 9.

For YOLOv4, the anchor box needs to be set in advance, so we adopted the k-means [25] clustering method for the dataset, producing 9 groups of anchor boxes, namely [(12,16), (19,36), (40,28)], [(36,75), (76,55), (72,146)], and [(142,110), (192,243), (459,401)]. Among these, the first three sets of anchor boxes correspond to the output of the 13 × 13 scale, and the middle three and the last three correspond to the output of the 26 × 26 and 52 × 52 scales, respectively. In addition, we conducted some preprocessing of the dataset before training [26].

### 5.2. Environment

The training and inference process of all the models in our study is completed on RTX3060, the deep learning framework used in our models is pytorch1.7, and the CUDA version is 11.2. The specific hyperparameter settings during model training in this study are shown in Table 1. In order to ensure the fairness of the experiment, the model involved in the ablation experiment uses the hyperparameters shown in Table 1.

### 5.3. Results and Analysis

In order to verify the effectiveness of the improvements made in this study, the improvements are gradually added on the basis of YOLOv4, and the final results are compared one by one for evaluation.

From Table 2, we see that mAP50 is 3.78% higher for our model than for YOLOv4, precision is improved by 3.55% and recall is improved by 9.91%. Each assessed indicator shows a corresponding improvement. This fully reflects the ability of NMYOLO to detect objects in the face of relatively complex situations, such as occlusion and overlap between infusion sets. Due to the addition of some detection and feature fusion modules, GFLOPs is increased by 1.88 frames while FPS is decreased by 2.79 frames when compared with YOLOv4. However, this still meets the frame rate requirements for video transmission.

In Table 3, the three indicators of recall, precision, and mAP50 of our model are 96.20%, 66.80%, and 95.21%, respectively, which are the best among the commonly used one-stage models that are compared. Meanwhile, the best value of mAP0.5:0.95 is 73.40%, which gets by YOLOv8m.The best value of mAP75 is 68.23% of YOLOv5m.

In Figure 10, when the recall is below 0.6, there are four categories with a precision close to 1, but after the recall is greater than 0.8, the precision decreases rapidly. In addition, the maximum value of each class recall is hardly close to 1. This shows that it is more difficult to detect objects in this dataset than it is to detect them correctly. It also shows that if we want to continue to improve mAP, we need to improve the recall of the model.

In order to facilitate the reader to directly and effectively see the improvements associated with our model, some of the predicted images of YOLOv4 are selected for comparison with some of the predicted images of NMYOLO in Figure 11.

The first column in Figure 11 is the original image, the second column is the result from YOLOv4 detection, and the third column is the result from NMYOLO detection. Among them, regarding the first line near the overlapping target scene, the original picture has three infusion bags and one infusion bottle; YOLOv4 detected two of these infusion bags and the one infusion bottle, thus missing one infusion bag, and NMYOLO detected all four targets, which shows that the NMYOLO has better detection efficacy in the face of overlapping occlusion targets. Regarding the second line of the distant target scene, NMYOLO and YOLOv4 both identify all three targets, but YOLOv4 wrongly identifies one corner of the box as a bottle, whereas there are no detection errors in the case of NMYOLO, indicating it has stronger stability in detecting distant targets. The third behavior is a complex scenario of overlapping multiple infusion bottles, and it is clear that NMYOLO has a higher detection rate for different placements and overlaps of infusion devices when faced with particularly complex infusion container scenarios.

## 6. Conclusions

To solve the difficult problem of detecting medical infusion containers under dense occlusion and in complex environment scenarios, we proposed the novel method NMYOLO. In this study, we enhance the depth of the neural network in the model by adding ASFF and CA, and improve the information fusion capability of the model. We also modify SPP to CSP-SPP to make the model obtain more information features, and use EIOU to make the model more stable. These improvements make NMYOLO have better detection. In addition, NMYOLO is shown to have better performance compared with other mainstream one-stage detection models. 

Although NMYOLO has served our purpose, we still need to discuss what its shortcomings are. The disadvantage of the proposed model is the reduction in the detection frame rate compared with YOLOv4, so one idea for subsequent improvement is to change the method of lightweight backbone or reduce some convolution modules that are not very important to reduce the number of parameters in the model. Moreover, we can replace some modules or change the architecture of the model to reduce the size of the model and ensure detection accuracy.

## Figures and Tables

**Figure 1 entropy-25-00275-f001:**
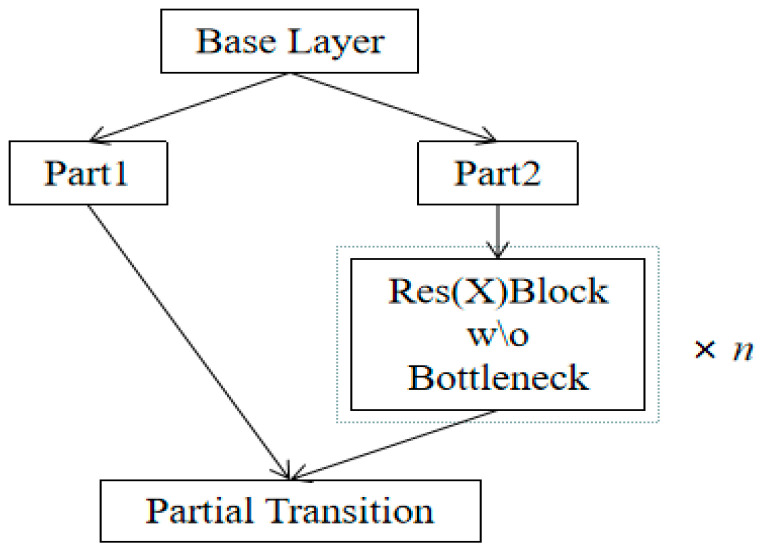
The structure of CSPResNe(X)t [19].

**Figure 2 entropy-25-00275-f002:**
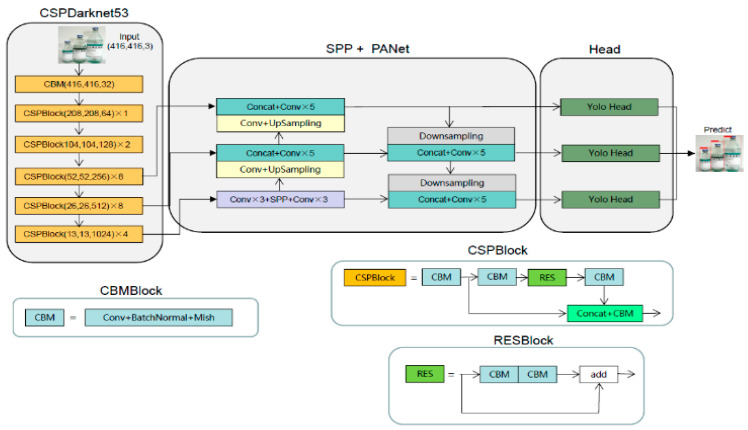
The structure of YOLOv4.

**Figure 3 entropy-25-00275-f003:**
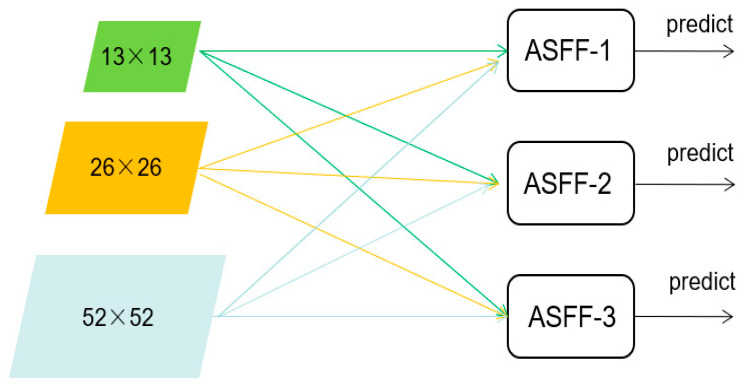
ASFF structure diagram.

**Figure 4 entropy-25-00275-f004:**
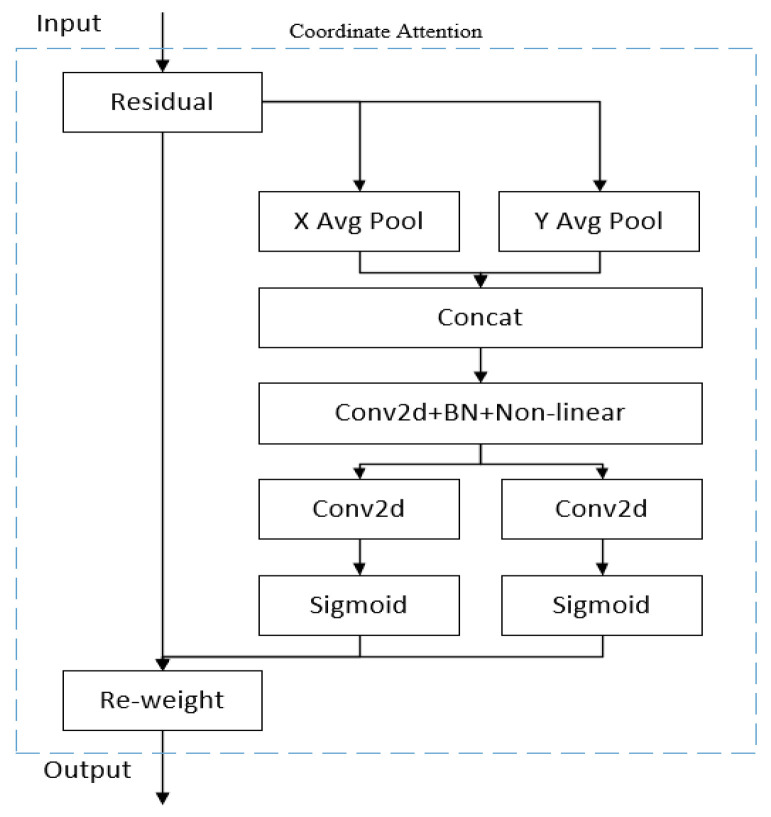
The structure of coordinate attention.

**Figure 5 entropy-25-00275-f005:**
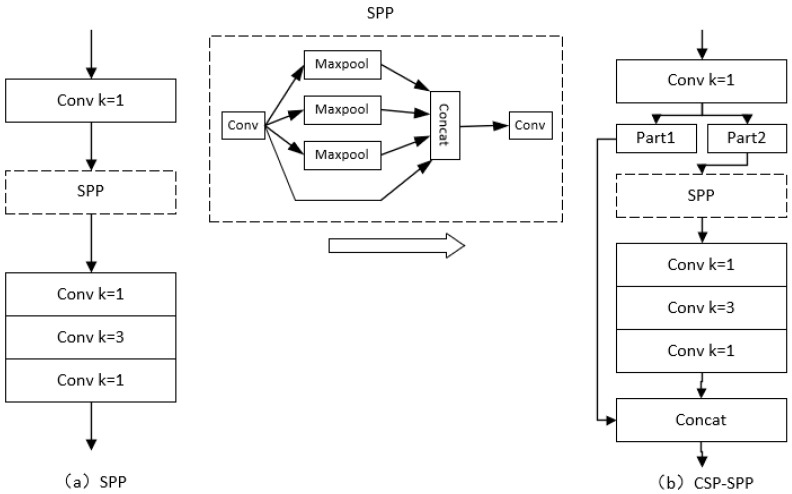
Improvement from SPP to CSP-SPP. (**a**) is the structure of SPP, (**b**) is the structure of CSP-SPP.

**Figure 6 entropy-25-00275-f006:**
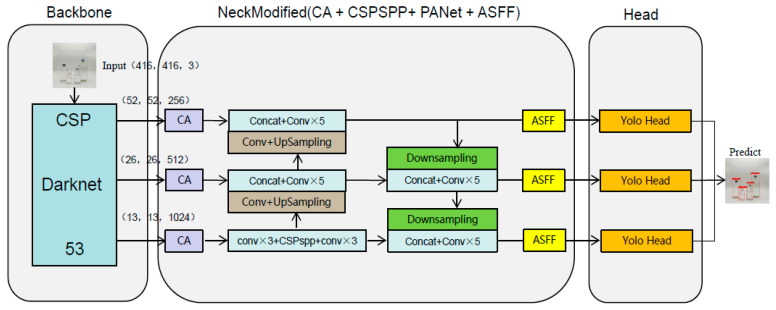
The structure of NMYOLO.

**Figure 7 entropy-25-00275-f007:**
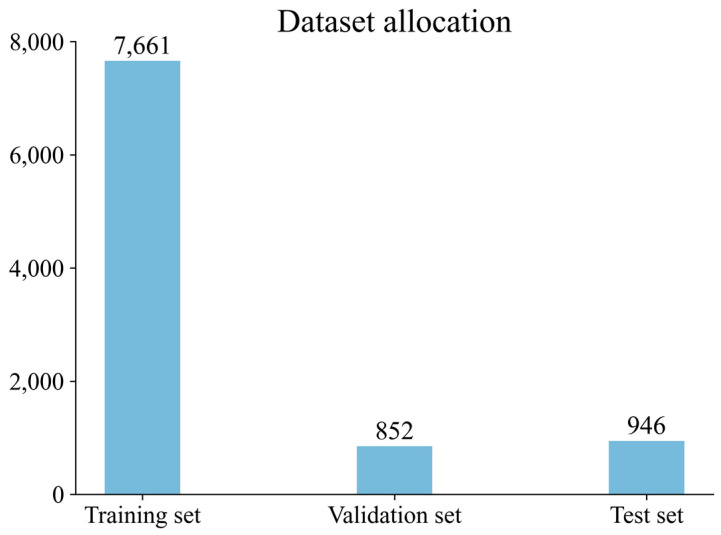
Dataset allocation.

**Figure 8 entropy-25-00275-f008:**
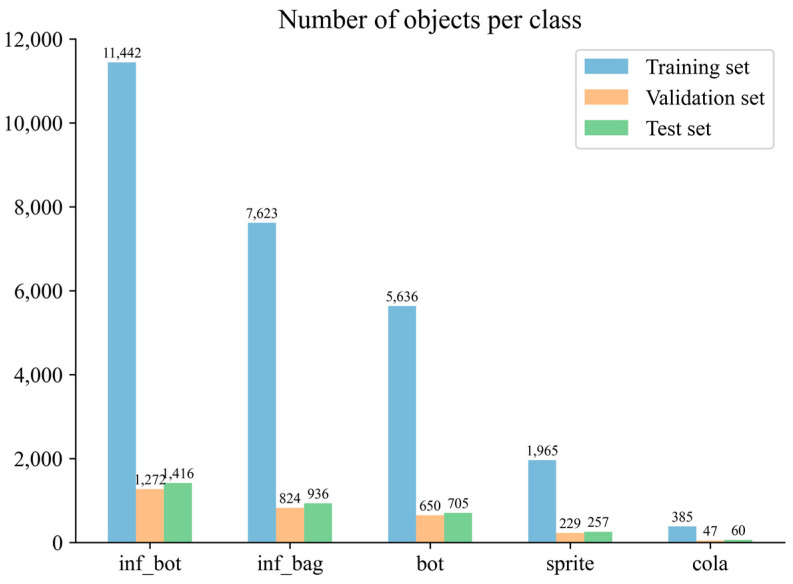
The number of objects per class.

**Figure 9 entropy-25-00275-f009:**
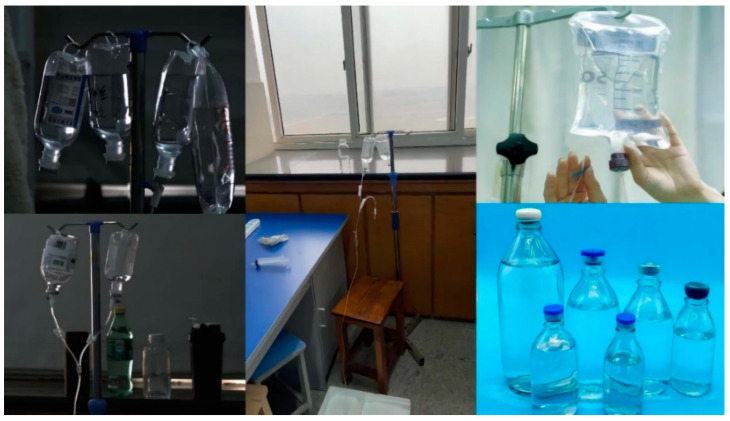
Some pictures from the dataset.

**Figure 10 entropy-25-00275-f010:**
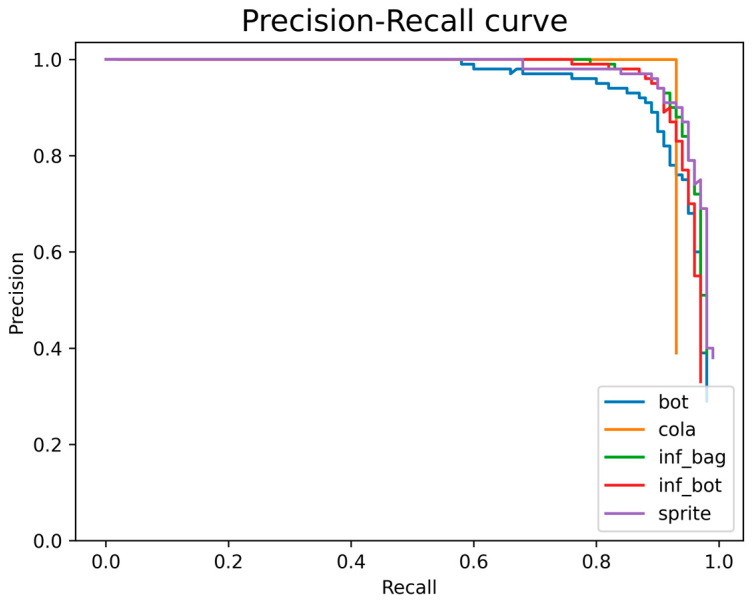
Precision–Recall curve of each class.

**Figure 11 entropy-25-00275-f011:**
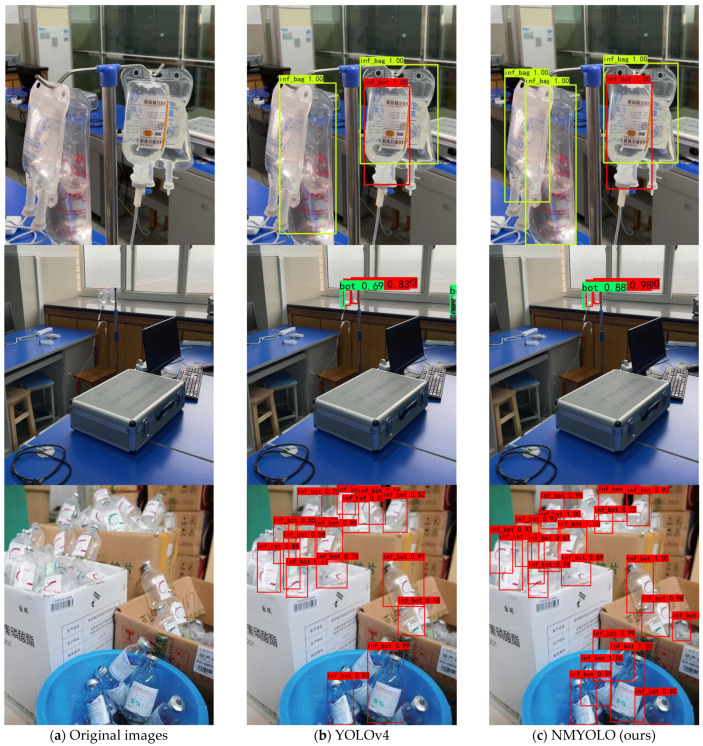
Original images and examples of detection results.

**Table 1 entropy-25-00275-t001:** List of hyperparameters.

Hyperparameters	Value
Optimizer	SGD
Learning rate	0.02
Momentum	0.937
Input shape	416 × 416
Weight decay	0.0005
Training Epochs	100
Cos_lr	True
Mosaic	True

**Table 2 entropy-25-00275-t002:** Comparison of the effects of ablation experiments. And ✗ means we will not add such a module to the baseline, ✓ means we will add it.

CA	ASFF	CSP-SPP	Loss	Precision	Recall	mAP50	GFLOPs	FPS
✗	✗	✗	CIoU	92.65	78.59	91.43	29.89	39.08
✓	✗	✗	CIoU	94.76	81.03	92.91	29.90	38.26
✓	✓	✗	CIoU	93.12	88.18	93.57	31.68	36.92
✓	✓	✓	CIoU	94.84	87.18	94.52	31.68	36.89
✓	✓	✓	EIoU	96.20	88.50	95.21	31.77	36.29

**Table 3 entropy-25-00275-t003:** The comparison between our model and other related one-stage models.

Methods	Input size	Precision	Recall	mAP0.5:0.95	mAP50	mAP75	FPS
SSD	416 × 416	88.76	74.38	42.70	83.58	39.36	/
YOLOv3	416 × 416	91.52	74.87	50.30	85.73	50.54	36.50
YOLOv3-spp	416 × 416	/	/	60.10	88.41	65.03	36.07
YOLOv4	416 × 416	92.65	78.59	62.00	91.43	61.03	39.08
YOLOv5m	416 × 416	93.28	81.77	66.10	91.97	68.23	72.03
YOLOv8m	416 × 416	91.60	87.60	73.40	94.40	/	69.04
YOLOX	416 × 416	93.86	85.12	66.50	93.67	68.16	56.03
YOLOv7	416 × 416	95.76	84.80	65.20	94.14	67.10	38.54
NMYOLO	416 × 416	96.20	88.50	66.80	95.21	67.90	36.29

## Data Availability

Not applicable.

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
