# Peer review of "An Infusion Containers Detection Method Based on YOLOv4 with Enhanced Image Feature Fusion"

_entropy, 2023, doi:10.3390/e25020275_

Round 1

Reviewer 1 Report

- Line 10, rephrase in more exact scientific manner, what is the difference between correctness and accuracy? You could have written, "satisfy clinical requirements"

- The introduction quickly goes into the related literature before precisely defining the problem and without any proper motivation to solve this problem.

- The paper lacks proper organization and structure. The introduction discusses the related works, but the related works section contains the materials and methods.

- What is the location of the feature extraction heads?

- It would be worthwhile to add appreciation of the used methods by citing relevant medical research that uses Yolo and Faster RCNN, see  Detection of K-complexes in EEG signals using deep transfer learning and YOLOv3. Cluster Comput (2022). https://doi.org/10.1007/s10586-022-03802-0 and Detection of K-complexes in EEG waveform images using faster R-CNN and deep transfer learning. BMC Med Inform Decis Mak 22, 297 (2022). https://doi.org/10.1186/s12911-022-02042-x

- Line 234, I don't understand why reference 24 is related!!

- The grammar and language of the manuscript need extensive review. 

- There is a need to report the precision, and the precision-recall curve. 

- The table of abbreviations is missing but it is required by the journal template.

Reviewer 2 Report

Section 2.3 is duplicated.

Reviewer 3 Report

In the introductory part I missed an explanation and emphasis on why an automatic infusion container detection is so important.

Nowadays I probably wouldn't mention almost 20 years old methods like AdaBoost, HOG, SIFT etc. , which are long outdated by CNNs.

I missed the reference to whether there are any other methods for detecting infusion containers and any comparison with these methods.

Highlight your contribution on the final detector, which parts are reproduced and which are your proposal.

Describe preprocessing steps on the databse.

As new versions of YOLO are available (version 7  and version 8), it would also be useful to include them in the comparison table.

Improve the discussion of the results and expand the summary considering weaknesses and strengths.

Round 2

Reviewer 1 Report

The authors answered my comments